# The Catecholaldehyde Hypothesis for the Pathogenesis of Catecholaminergic Neurodegeneration: What We Know and What We Do Not Know

**DOI:** 10.3390/ijms22115999

**Published:** 2021-06-01

**Authors:** David S. Goldstein

**Affiliations:** Autonomic Medicine Section, Clinical Neurosciences Program, Division of Intramural Research, National Institute of Neurological Disorders and Stroke, National Institutes of Health, Bethesda, MD 20892, USA; goldsteind@ninds.nih.gov; Tel.: +301-675-1110; Fax: +301-402-0180

**Keywords:** catecholaldehyde, DOPAL, autotoxicity, monoamine oxidase, dopamine, alpha-synuclein

## Abstract

3,4-Dihydroxyphenylacetaldehyde (DOPAL) is the focus of the catecholaldehyde hypothesis for the pathogenesis of Parkinson’s disease and other Lewy body diseases. The catecholaldehyde is produced via oxidative deamination catalyzed by monoamine oxidase (MAO) acting on cytoplasmic dopamine. DOPAL is autotoxic, in that it can harm the same cells in which it is produced. Normally, DOPAL is detoxified by aldehyde dehydrogenase (ALDH)-mediated conversion to 3,4-dihydroxyphenylacetic acid (DOPAC), which rapidly exits the neurons. Genetic, environmental, or drug-induced manipulations of ALDH that build up DOPAL promote catecholaminergic neurodegeneration. A concept derived from the catecholaldehyde hypothesis imputes deleterious interactions between DOPAL and the protein alpha-synuclein (αS), a major component of Lewy bodies. DOPAL potently oligomerizes αS, and αS oligomers impede vesicular and mitochondrial functions, shifting the fate of cytoplasmic dopamine toward the MAO-catalyzed formation of DOPAL—destabilizing vicious cycles. Direct and indirect effects of DOPAL and of DOPAL-induced misfolded proteins could “freeze” intraneuronal reactions, plasticity of which is required for neuronal homeostasis. The extent to which DOPAL toxicity is mediated by interactions with αS, and vice versa, is poorly understood. Because of numerous secondary effects such as augmented spontaneous oxidation of dopamine by MAO inhibition, there has been insufficient testing of the catecholaldehyde hypothesis in animal models. The clinical pathophysiological significance of genetics, emotional stress, environmental agents, and interactions with numerous proteins relevant to the catecholaldehyde hypothesis are matters for future research. The imposing complexity of intraneuronal catecholamine metabolism seems to require a computational modeling approach to elucidate clinical pathogenetic mechanisms and devise pathophysiology-based, individualized treatments.

## 1. Introduction

The “catecholaldehyde hypothesis” offers a unitary answer to four key questions about the pathogenesis of Parkinson’s disease (PD) and related Lewy body diseases: (1) only a very small percent of neurons are catecholaminergic. What makes them so susceptible, such that they are relatively selectively lost? (2) How do generalized abnormalities such as mutations and exposure to environmental toxins lead to specific loss of catecholaminergic neurons? (3) Why does the protein alpha-synuclein (αS) tend to precipitate in catecholaminergic neurons in Lewy body diseases? And (4) Why are Lewy body diseases aging related?

3,4-Dihydroxyphenylacetaldehyde (DOPAL) is the focus of the catecholaldehyde hypothesis. DOPAL is formed from dopamine in the neuronal cytoplasm via enzymatic deamination catalyzed monoamine oxidase (MAO) (Figure 1). Despite the central position of DOPAL in the disposition of endogenous dopamine, the literature on DOPAL (based on listings in PubMed) consists of only approximately 130 articles, in contrast with approximately 170,000 for dopamine. The starting point in presenting the catecholaldehyde hypothesis is the role of MAO in DOPAL production [1].

## 2. MAO and Intraneuronal Metabolism of Catecholamines

In 1928, Mary L.C. Hare described an enzyme in the liver that oxidized and de-aminated the dietary amine tyramine and resulted in the formation of ammonia [2]. She noted that the amount of ammonia generated by this amine oxidase was one-half that of the oxygen consumed. Subsequently, Kohn explained this in terms of the formation of hydrogen peroxide (which is then metabolized by catalase) and an aldehyde, 4-hydroxyphenylacetaldehyde [3]. Demonstration that the catecholamine adrenaline and tyramine competed for the same enzyme led to abandonment of the notion of a separate “adrenaline oxidase”. 

After Julius Axelrod showed that the major route of exogenously administered catecholamine is O-methylation catalyzed by catechol-O-methyltransferase (COMT) [4], routes of inactivation of endogenous catecholamines were unclear. Eventually, it was recognized that in contrast with exogenously administered cate-cholamines, catecholamines produced within neurons (norepinephrine and dopamine) are metabolized mainly by MAO [5,6]. 

Within cells MAO is localized to the outer mitochondrial membrane [7]. This localization is potentially important, because recent evidence suggests that in contrast with inferences from test tube experiments, in catecholaminergic cells, an effect of enzymatic oxidative deamination of dopamine is not to generate hydrogen peroxide in the cytoplasm but to stimulate the mitochondrial electron transport system [8].

Youdim, Sandler, and colleagues obtained evidence for multiple isoforms of MAO [9,10], which came to be called MAO-A and MAO-B. In 1968 Johnston noted that MAO-A is efficiently antagonized by clorgyline [11]. The drug deprenyl (the levo-rotatory stereoisomer of selegiline) was found to antagonize MAO-B [12]. The genes encoding MAO-A and MAO-B occur adjacent to each other on the X-chromosome [13,14]. 

MAO-A has been thought to be characterized by high affinity for norepinephrine and serotonin and sensitivity to inhibition by clorgyline and MAO-B by high affinity for phenylethylamine and benzylamine and sensitivity to inhibition by deprenyl [15]. It has also been thought that dopamine is equally susceptible to metabolism by MAO-A and MAO-B [16]. Although it has long been recognized that dopamine is more readily metabolized by MAO than is norepinephrine or epinephrine, the relative susceptibilities of catecholamines to MAO-A and MAO-B have not yet been assessed formally.

Although most of MAO activity in the brain is of the B type, MAO-A figures prominently in the oxidative deamination of striatal dopamine [17,18,19,20,21]. Based on mRNA contents, MAO-A is expressed in noradrenergic and dopaminergic cell groups, with relatively little MAO-B expression [22]. Substantia nigra dopaminergic neurons of mice express both MAO-A and MAO-B, with a predominance of MAO-A [8]. Sympathetic nerves contain mainly MAO-A [23]. 

Administration of drugs that are selective MAO-B inhibitors in vitro can inhibit MAO-A in vivo [24,25,26]. Bases for this phenomenon are unclear but seem to be related to the chronicity of dosing [27]. In humans, treatment with the MAO-B inhibitors deprenyl or rasagiline decreases plasma levels of 3,4-dihydroxyphenylacetic acid (DOPAC) and 3,4-dihydroxyphenylglycol (DHPG), the respective main intra-neuronal metabolites of dopamine and norepinephrine [24,28] (Figure 1, Figure 2 and Figure 3).

## 3. DOPAL Toxicity

After Blaschko’s expression of concern in 1952 that aldehydes produced from MAO-catalyzed oxidative deamination of catecholamines might be toxic [29], about 40 years elapsed before the first evidence of catecholaldehyde toxicity was reported by Mattammal et al. [30]. Applying a gas chromatography-mass spectrometry method, they noted the presence of DOPAL in post-mortem substantia nigra tissue from patients with PD but not in control subjects [31]. In the same study, they reported that incubation of dopamine with MAO-B resulted in 4–5-fold increases in covalent binding to DNA, bovine serum albumin, and microtubular protein. This finding presaged the relatively recent discovery of DOPAL-mediated oli-gomerization and quinone adduct formation (“quinonization”) of numerous proteins in catecholaminergic cells [32]. 

Mattammal et al. [30] also reported that (1) DOPAL is taken up into striatal synaptosomes via the cell membrane dopamine transporter (DAT); (2) DOPAL concentration- and time-dependently releases DA from synaptosomes, due to damaging the synaptosomal membranes; (3) in rat pheochromocytoma PC12 cells differentiated with nerve growth factor, DOPAL concen-tration-dependently depletes cell contents of dopamine, DOPAC, and homovanillic acid (HVA, the end product of dopamine metabolism) and kills cells based on both cell counts and release of lactate dehydrogenase into the medium; (4) in primary cultures of ventral midbrain from rat embryos, at DOPAL concentrations of 7.5–20 µM, there are reductions in dopamine uptake without a reduction in the number of tyrosine hydroxylase (TH)-positive cells, and at higher concentrations, there is cellular loss; and (5) surviving TH-positive cells show rounded cell bodies and fragmented fiber networks. The authors concluded that DOPAL may be an endogenous toxin in catecholaminergic cells—an autotoxin, referring to an endogenous substance that can harm the same cells in which it is produced [33].

Studies soon afterward by Burke and colleagues focused on DOPAL as a pathogenetic factor in PD. Kristal et al. reported that DOPAL targets mitochondria. In energetically compromised mitochondria DOPAL induces mitochondrial permeability transition, a trigger for cell death by necrosis or apoptosis [34]. Mitochondrial pore opening permits numerous solutes to diffuse from the cytoplasm across the mitochondrial inner membrane, depolarizing the mitochondria, uncoupling oxidative phosphorylation, and impeding generation of ATP that required for vital cellular processes. In this regard, DOPAL is far more potent than dopamine [35]. 

Li et al. reported that DOPAL reacts with hydrogen peroxide to generate extremely toxic hydroxyl radicals [36]. The relevance of this finding to pathophysiology is unclear, because hydrogen peroxide is broken down by catalase and because within cells MAO acting on dopamine may not actually increase cytoplasmic hydrogen peroxide levels [8].

Burke et al. examined the in vivo toxicity of exogenously administered DOPAL in rat substantia nigra and ventral tegmental area. At 5 days post-surgery, DOPAL at con-centrations as low as 100 ng was found to be toxic selectively to nigral dopaminergic neurons [37]. Several years later, another in vivo animal study of exogenously administered DOPAL in rats reported that DOPAL evokes neuronal loss in the substantia nigra pars compacta and loss of striatal dopaminergic terminals and produces rotational asymmetry as is seen in other PD animal models [38].

## 4. Toxicity from Spontaneous Oxidation of Cytoplasmic Dopamine

Fornstedt and Carlsson discovered that MAO inhibition increases the formation of 5-S-cysteinyldopamine (Cys-DA) in guinea pig brain [39]. This spurred a line of research implicating dopamine itself as an autotoxin, based on oxidation of dopamine to dopamine-quinone and then a variety of distal oxidation products [40,41,42,43,44,45,46,47,48,49,50,51,52,53,54,55,56,57,58,59,60,61,62,63,64,65,66] (Figure 1 and Figure 2). Some of these are known to be neurotoxic, such as aminochrome [47,67,68], Cys-DA [66,69], and isoquinolines [70,71]. These products of spontaneous dopamine oxidation have in common that they are toxic to mitochondria [54]. 

## 5. DOPAL Detoxification by Aldehyde Dehydrogenase (ALDH)

Blaschko theorized that enzymatic metabolism via mitochondria detoxifies the aldehydes produced by amine oxidases acting on monoamines. He wrote, “…the amine oxidase reaction results in the formation of two toxic metabolites, both of which are removed by reactions which take place in the mitochondria. One of these metabolites is ammonia which is removed in the mitochondria by urea synthesis, the other is an aldehyde which is further metabolised in reactions known to be linked with the cytochrome-cytochrome oxidase system. The location of amine oxidase in the mitochondria thus ensures that these two end products of the enzymatic reaction are rendered harmless not far from the site where they are formed” [29]. In 1957 he wrote, “Oxidative deamination by amine oxidases leads to the formation of an aldehyde, but the aldehyde is usually further oxidized to the corresponding carboxylic acid.” [72].

DOPAL is converted to the acid DOPAC via aldehyde dehydrogenase (ALDH), and the DOPAC produced rapidly exits the cells. Thus, in PC12 cells DOPAC is extruded actively via a sulfonylurea-sensitive transporter [73]. One may conceptualize that MAO keeps cytoplasmic dopamine levels low, and ALDH in series keeps DOPAL levels low. 

In the 1960s Youdim and Sandler published that reserpine inhibits ALDH [74]. To explain this phenomenon they wrote, “A facilitated metabolism of catecholamine, giving rise in its turn to an increased production of aldehyde, may cause substrate inhibition of aldehyde dehydrogenase.” Subsequent experimental evidence has supported the concept that DOPAL can inhibit its own metabolism by ALDH (Figure 4). There is a linear negative relationship between the percent of control ALDH activity and DOPAL concentrations in the range of 0–60 µM [75].

ALDH is an important determinant of intracellular DOPAL levels. Mice with double knockout of the genes encoding the mitochondrial and cytosolic forms of ALDH have DOPAL buildup [76]. The fungicide benomyl inhibits ALDH and also results in DOPAL accumulation [77]. Farm chemicals inhibiting ALDH may contribute to the incidence of PD [78,79,80]. In humans it has been reported that decreased ALDHA1A gene expression in blood is part of a “molecular signature” that can identify early PD [81]. ALDH1A1 gene expression and protein are decreased in substantia nigra specimens in patients with PD [82,83]. 

The mitochondrial complex 1 inhibitor rotenone indirectly decreases ALDH activity [84] by interfering with its co-factor NAD^+^, and rotenone therefore increases endogenous DOPAL production in PC12 cells [85]. Predictably, ALDH2 activation is protective in cellular and animal models of rotenone-induced neurotoxicity [86].

Graves et al. [8] recently advanced the concept that rather than MAO resulting in hydrogen peroxide production in the cytoplasm, because of the positioning of MAO in the outer mitochondrial membrane electrons are shuttled through the intermembrane space to increase activity of the electron transport cascade leading to ATP production. The model does not mention the consequences of DOPAL production or that generation of NAD+ is required for ALDH to detoxify DOPAL.

It should be noted that both mitochondrial and cytosolic ALDH metabolize DOPAL. The human genome contains 19 genes for ALDH [87], and it is unclear which of these are mitochondrial and which cytosolic.

The lipid peroxidation products 4-hydroxynonenal and malondialdehyde can build up DOPAL via inhibition of ALDH [88,89]. Conversely, DOPAL might promote buildup of the lipid peroxidation products by interfereing with their metabolism. 

## 6. Vesicular Uptake as a Detoxification Mechanism

Active uptake of cytoplasmic dopamine into vesicles via the vesicular monoamine transporter (VMAT) is not only required for dopaminergic neurotransmission but also serves as a detoxification mechanism [90,91,92,93] (Figure 2). This includes autotoxicity exerted by dopamine itself [41]. Animals with reduced activity of the type 2 VMAT have evidence of progressive nigrostriatal and locus ceruleus neurodegeneration [94,95], while increased VMAT activity is neuroprotective [58,96]. Unexpectedly, mice with genetically determined very low VMAT2 activity do not have evidence of DOPAL buildup [97]. These animals have elevated tissue DOPAC/DOPAL ratios, suggesting differential survival based on increased DOPAL detoxification by ALDH. 

## 7. The Double Hit Concept

For a given rate of cytoplasmic dopamine synthesis, DOPAL levels are determined by dopamine uptake into vesicles and DOPAL metabolism (Figure 2). Although a small amount of DOPAL is metabolized to 3,4-dihydroxyphenylethanol (DOPET) by aldehyde/aldose reductase (AR), the main enzymatic fate of cytoplasmic DOPAL is metabolism to DOPAC via ALDH. Accordingly, blockade by reserpine of vesicular uptake increases endogenous DOPAL levels in PC12 cells, and concurrent inhibition of ALDH increases DOPAL levels further [98]. Post-mortem neurochemical evidence for a “double hit” of a vesicular storage defect with decreased ALDH activity has been reported in putamen tissue from patients with PD [99].

## 8. DOPAL-Protein Interactions, with Emphasis on Alpha-Synuclein

DOPAL interacts with numerous intra-cellular proteins related to catecholamine production, storage, recycling, and metabolism (Figure 4). As noted above, DOPAL interferes with synaptosomal uptake and retention of tracer-labelled dopamine [30]. DOPAL covalently binds to and inhibits the activity of TH [100], the rate-limiting enzyme in catecholamine biosynthesis [101]. The ability of DOPAL to do so depends on both the catechol and aldehyde moieties [102]. DOPAL also forms quinoprotein adducts with and inhibits the activity of LAAAD [32] and may decrease its own metabolism by ALDH [75].

In 1997, the first genotypic abnormality producing familial PD was identified—A53T mutation of the gene encoding αS [103]. In the same year, Lewy bodies, a histopathologic hallmark of idiopathic PD, were found to contain abundant αS [104]. Since then the view has evolved that PD as typically encountered clinically is a form of synucleinopathy. Other synucleinopathies include multiple system atrophy (MSA), in which αS is deposited in glial cytoplasmic inclusions in the brain [105]; dementia with Lewy bodies [106]; and pure autonomic failure (PAF) [107,108]. 

Since then it has been suggested that dopamine oxidation products contribute to the pathogenesis of PD by interactions with αS [33,65,109,110]. There are two general routes by which this could happen. First, dopamine can promote the formation of αS oligomers [48,56,111]. Non-enzymatic oxidation of dopamine results in formation of dopamine-quinone (DA-Q), and oxidized dopamine interacts with αS [44]. Moreover, aminochrome and 5,6-dihydroxyindole, which are products of dopamine oxidation, can oligomerize αS [112,113,114]. 

Second, DOPAL can oxidize spontaneously to DOPAL-quinone (DOPAL-Q) [32,115,116] (Figure 5 and Figure 6). Moreover, the enzyme cyclooxygenase-2 can oxidize DOPAL enzymatically [116]. DOPAL oligomerizes aS [117], probably via DOPAL-Q [32], and oligomerized aS seems to be pathogenic [118,119,120]. Divalent metal cations—especially Cu(II)—augment DOPAL-induced aS oligomerization [121]. DOPAL also forms covalent quinone adducts with (“quinonizes”) many other PD-related proteins [32]. Quinonization may interfere with the functions of these proteins and thereby with numerous intracellular processes.

DOPAL-induced αS oligomers impede vesicular functions [123], which could set the stage for a vicious cycle. DOPAL-induced αS oligomers also have been reported to inhibit mitochondrial oxygen consumption and decrease the mitochondrial membrane potential [124]. Divalent metal cations—especially Cu(II)—augment DOPAL-induced oligomerization of αS [121], while anti-oxidation with reduced glutathione, ascorbic acid, or N-acetylcysteine (NAC) attenuates the oligomerization [32,115,125]. 

DOPAL-induced oligomerization of αS has been proposed to reflect condensation of two DOPAL molecules in a dicatechol pyrrole lysine adduct, followed by formation of isoindole linkages [126,127]. Superoxide radical drives this process [128]. The reaction of DOPAL with lysine residues results in production of reactive oxygen species [126]. Since superoxide is also generated when DOPAL oxidizes, this is another potential vicious cycle. A recent report supported the catecholaldehyde hypothesis, in that Schiff base adducts between DOPAL and the amines rasagiline or aminoindan were found to inhibit DOPAL-induced αS aggregation and toxicity [129].

DOPAL also evokes the formation of quinone adducts with many proteins (“quinonization”) relevant to catecholaminergic functions, including TH, L-aromatic-amino-acid-decarboxylase (LAAAD), and the type 2 VMAT [32]. DOPAL also oligomerizes and quinonizes ubiquitin [32], suggesting that DOPAL may interfere with the proteasomal disposition of other misfolded proteins. 

The literature on DA oxidation and synucleinopathy has generally overlooked DOPAL [48,64,65,110,130,131,132], and the literature on DOPAL and synucleinopathy has generally overlooked DA-Q [32,115,117,133]. In the few studies where DOPAL and dopamine have been compared directly in terms of oligomerizing α-syn, DOPAL has been found to be far more potent [32,117,121,122]. A recent study found that DOPAL is not only more potent than dopamine in oligomerizing αS but also that DOPAL quinonizes αS [122]. Even in the setting of dopamine oxidation evoked by Cu(II) or tyrosinase, dopamine does not quinonize αS (Figure 5).

The differences in potencies of DOPAL and dopamine in oligomerizing and quinonizing αS may be explained by their different chemical structures [134]. Dopamine has a terminal amine group, while DOPAL has a reactive aldehyde group that can bind covalently to lysine residues [125], which are abundant in the αS molecule [115,123,134]. Thus, occupation of lysine residues completely prevents both the oligomerization and quinonization of αS by DOPAL [32]. 

The discovery that DOPAL induces quinonization of numerous intracellular proteins is relatively new, and there is scant literature about how this happens. N-acetylcysteine (NAC) inhibits DOPAL-induced quinonization of αS [32,122], consistent with dependence of the quinonization on oxidation of DOPAL to DOPAL-Q. Whether quinonization is a step in the oligomerization of αS remains unknown.

The extent of quinonization of αS by Cu(II)+DOPAL is about twice as great with the A53T mutant form of αS than with human wild-type αS [122]. This finding suggests that, at least in the setting of Cu(II), neurons expressing mutant forms of αS may be more vulnerable to DOPAL-induced quinonization than neurons expressing wild-type αS [122]. So far there has been no demonstration that DOPAL-induced cytotoxicity is mediated by protein quinonization. 

Braak’s “gut first” concept states that “a putative environmental pathogen capable of passing the gastric epithelial lining might induce αS misfolding and aggregation in specific cell types of the submucosal plexus and reach the brain via a consecutive series of projection neurons” [135]. Almost half of the synthesis and metabolism of dopamine in the body takes place in non-neuronal cells of the gut [136]. One may speculate that DOPAL produced locally from abundant non-neuronal dopamine reacts with αS to induce a pathogenic cascade.

A recent study involved DOPAL injection into the vagus nerve, to help understand the frequent association of autonomic failure with PD [137]. DOPAL-treated rats had evidence of baroreflex dysfunction, orthostatic hypotension, and time-dependent associated changes in αS monomers/trimers. The data suggest the plausibility of “body-first” subtype of PD that features autonomic abnormalities [138].

αS may also interact with DOPAL by increasing activities of MAO-A or MAO-B, which would be expected to build up DOPAL for given rates of production of dopamine in the cytoplasm. Kang et al. recently reported that αS binds to and directly increases activity of MAO-B [139]. This in turn triggers asparagine endopeptidase and cleavage of αS at N103. The cleavage is then thought to lead to dopaminergic neurodegeneration. The authors found that DOPAL strongly activates asparagine endopeptidase, and the N103 fragment of αS binds and activates MAO-B, which could result in a lethal positive feedback loop. Jia et al. recently reported that αS up-regulates MAO-A [140].

Brain-derived neurotrophic factor (BDNF)/tyrosine kinase B (TrkB) signaling is required for the survival of dopaminergic neurons. Kang et al. reported that αS interacts with TrkB receptors and inhibits BDNF/TrkB signaling, leading to dopaminergic neuronal death. DOPAL stimulates this harmful interaction, and rasagiline prevents the αS-induced dopaminergic neuronal death and restores motor functions [141].

As scanty as the literature is about DOPAL-related neurotoxicity, there is even less literature about DOPEGAL, which is the catecholaldehyde produced by the action of MAO on norepinephrine (Figure 3). An early neuropathological feature of Alzheimer’s disease is deposition of the protein Tau in neurons of the pontine locus ceruleus, which is the main source of norepinephrine in the brain. Kang et al. recently reported that DOPEGAL activates asparagine endopeptidase, which cleaves Tau and results in aggregation- and propagation-prone forms of the protein, leading to both locus ceruleus degeneration and spread of Tau pathology [142]. The same group has reported that DOPAL induces activation of asparagine endopeptidase [139]. Intradermal DOPEGAL injection has also been reported to produce mechanical hyperalgesia in an animal model of alcoholic peripheral neuropathy [143].

## 9. Stress and the Catecholaldehyde Hypothesis

Catecholaminergic systems operate differently from other neurotransmitter systems in that the neurotransmission is relatively slow [144], and the sites of neurotransmitter release are not necessarily synaptic and can occur at varicosities along widely arborizing axons [145,146]. Catecholamine neurons behave as if continuously in “idle,” analogous to a bank robber’s getaway car [147]. When needed, on top of this idling, there is augmented release in response to and even in anticipation of emergencies and activities of daily life. 

One may ask what the value is of “leaky” vesicles that cause a high rate of turnover of catecholamines even without exocytotic release. Extending on the automotive analogy, an explanation offered by Eisenhofer is based on “gearing down” [148]. In the human heart about 3/4 of norepinephrine turnover is due to intraneuronal metabolism of norepinephrine leaking from storage vesicles. Norepinephrine stores are in a highly dynamic equilibrium with the cytoplasm, and passive leakage is balanced by active uptake via the VMAT. Having leaky stores may gear down the requirement for increases in catecholamine synthesis to match increases in catecholamine release and neurons with a capacity for more extended range of sustainable release rates in response to stress than would otherwise be possible.

Activation of the “central stress system,” conceptualized to be embedded in the central autonomic network [149,150], increases exocytotic release of catecholamines in the brain and periphery [151,152,153]. Most of neurotransmitter catecholamines are recycled by neuronal reuptake [154,155] mediated by the DAT and NET. In essence, stress therefore results in a shift from vesicular stores to the cytoplasm, where the catecholamines can be oxidized by MAO. Repeated episodes of distress might then promote neuronal injury via autotoxicity.

Although direct evidence is lacking, the results of several studies fit with this notion. In rats, repeated immobilization, which activates catecholaminergic neurons inside and outside the brain [156,157,158,159], reduces the numbers of substantia nigra dopaminergic and locus ceruleus noradrenergic neurons [160], as in PD [161]. Exposure to multiple, random unpredictable stressors exacerbates the loss of TH-positive neurons in the substantia nigra that is evoked by the neurotoxin 6-hydroxydopamine [162]. In mice, chronic exposure to a mild stress paradigm augments neurotoxin-related neurodegeneration [163]. Chronic restraint (a severe stressor) promotes rotenone-induced neurotoxic effects in the brain, including loss of substantia nigra TH-positive cells and reduced striatal concentrations of dopamine [164]. In this study, rotenone alone did not cause overt nigral neuronal loss.

## 10. MAO Inhibitor Trials and the Catecholaldehyde Hypothesis

Since MAO is required for DOPAL formation, a seemingly straightforward test of the catecholaldehyde hypothesis would be to assess whether MAO inhibitors slow the progression of PD; however, results of large multi-center trials of the MAO-B inhibitors deprenyl and rasagiline failed to demonstrate efficacy convincingly [165,166,167,168,169]. 

There are two potential explanations for this failure. One is that the subjects in these clinical trials already had symptomatic PD, and the neurodegenerative process might already have been advanced by the time their symptoms manifested clinically. There seems to be a long preclinical period during which catecholaminergic neurons are dysfunctional—we call this the “sick-before-dead” phenomenon [170]. In at-risk individuals, cerebrospinal fluid indices of central dopamine deficiency predict the later development of PD [171]. 

Second, as inspection of the steps in Figure 1 would predict, MAO inhibitors increase the spontaneous oxidation of cytoplasmic dopamine, as indicated by increased levels of Cys-DA [28,39]. We call this the “MAOI tradeoff.” The catecholaldehyde hypothesis has not yet been put to a correct test in humans. 

The dietary supplements NAC and hydroxytyrosol (synonymous with DOPET) do not interfere with the ability of MAO-B inhibitors to decrease endogenous DOPAL production, yet they attenuate the MAO inhibitor-induced increase in Cys-DA [172,173]. A reasonable strategy would be to combine NAC or hydroxytyrosol with an MAO inhibitor. A trial of oral and intravenous NAC reported retardation in the progression of symptoms of PD and of the striatal dopaminergic lesion [174,175]. NAC also mitigates DOPAL-induced αS oligomerization, probably via decreasing the spontaneous oxidation of DOPAL to form DOPAL-Q [32].

Ideally, such a trial would involve patients with early disease or even people at risk for PD who have biomarkers of catecholaminergic neurodegeneration but without motor signs. Cardiac sympathetic neuroimaging evidence of myocardial noradrenergic deficiency and low cerebrospinal fluid levels of DOPA and DOPAC predict PD in at-risk individuals [176]. In patients with PD, the severity of the cardiac noradrenergic lesion progresses over time [170], and PAF can evolve into PD, DLB, or both [177]. By combining biomarkers of catecholaminergic dysfunction in extant neurons—the sick-but-not-dead phenomenon [178,179]—with biomarkers of deposition of αS in sympathetic noradrenergic nerves [180], an enriched enough population might be identified for efficient testing of the catecholaldehyde hypothesis.

## 11. Gaps in Knowledge and Goals for the Future

Even cursory inspection of the concept diagrams in Figure 2, Figure 3 and Figure 4 and 6 bring to mind gaps in knowledge and challenges that future research can address. This section frames the issues in terms of as yet unanswered—but answerable—questions. 

(1)Regarding the metabolic fate of cytoplasmic dopamine, what proportions of cyto-plasmic dopamine undergo spontaneous vs. enzymatic oxidation? Why do MAO-B-selective drugs decrease levels of catecholamine metabolites thought to be produced within neurons that express MAO-A?(2)Regarding DOPAL-induced quinonization of intracellular proteins, what are the functional consequences for the proteins involved with reactions in catechola-minergic neurons? Does DOPAL quinonize intracellular proteins including αS in vivo in sympathetic noradrenergic neurons, such as in skin biopsies? Do Lewy bodies contain DOPAL-quinonized proteins?(3)Regarding DOPAL–αS interactions and disease pathogenesis, what are the relative contributions of DOPAL and alpha-synucleinopathy to catecholaminergic neuro-degeneration? In the absence of synucleinopathy is DOPAL accumulation patho-genic or a non-pathogenic biomarker? Do DOPAL and αS ascend together in the periphery in “body-first” PD or PAF?(4)At this point, there is insufficient literature about whether catecholaldehydes affect neurogenesis or neuroinflammation.(5)Diagnostic and therapeutic applications of the catecholaldehyde hypothesis would be especially valuable in preclinical disease. Cerebrospinal fluid indices of central catecholamine deficiency provide predictive biomarkers in individuals at risk of developing PD [171] and might be especially valuable in identifying appropriate candidates for clinical treatment or prevention trials.

Because of the daunting complexity of intraneuronal catecholamine metabolism and the numerous potential sites of DOPAL-related abnormalities in Lewy body diseases, a computational modeling approach seems necessary to identify patterns of altered reac-tions, predict the course of disease, and incite individualized treatment or preventive measures. Although such modeling has identified specific sites of abnormal catecholamine synthesis, storage, and metabolism in Lewy body diseases [178], efforts to apply computational modeling so far have not incorporated homeostasis, allostatic load, genetics, aging, stress, and autotoxicity.

## 12. Conclusions

According to the catecholaldehyde hypothesis, the answers to the four questions that began this review are as follows. Catecholaminergic are especially susceptible, because DOPAL formed within catecholaminergic neurons acts as an autotoxin. (2) Mutations and exposure to environmental toxins can lead to specific loss of catecholaminergic neurons because these abnormalities promote the formation or spontaneous oxidation of DOPAL, interfere with its detoxification, or enhance DOPAL-induced protein misfolding or mi-tochondrial dysfunction. αS tends to precipitate in catecholaminergic neurons in Lewy body diseases, because DOPAL potently oligomerizes, quinonizes, and aggregates αS. Lewy body diseases are aging related, because the harmful effects of DOPAL or DOPAL–αS interactions are cumulative. During a long preclinical period, compensatory adjust-ments can maintain vesicular stores of catecholamine neurotransmitters, but eventually there is a failure of neuronal homeostasis and induction of multiple lethal positive feedback loops.

Catecholamine autotoxicity has long been considered a possible pathogenetic mechanism of catecholaminergic neurodegeneration in PD, based on the well-known tendency of dopamine to oxidize spontaneously, resulting in the formation of several potentially toxic oxidation products. The catecholaldehyde hypothesis, which is based on enzymatic rather than spontaneous oxidation of cytoplasmic dopamine, is relatively new. It is hoped that this review will spur interest in further research on DOPAL-induced catecholamine autotoxicity for understanding mechanisms, identifying biomarkers, and devising novel treatment and prevention strategies for diseases that are posing an in-creasing public health burden as populations age. 

## Figures and Tables

**Figure 1 ijms-22-05999-f001:**
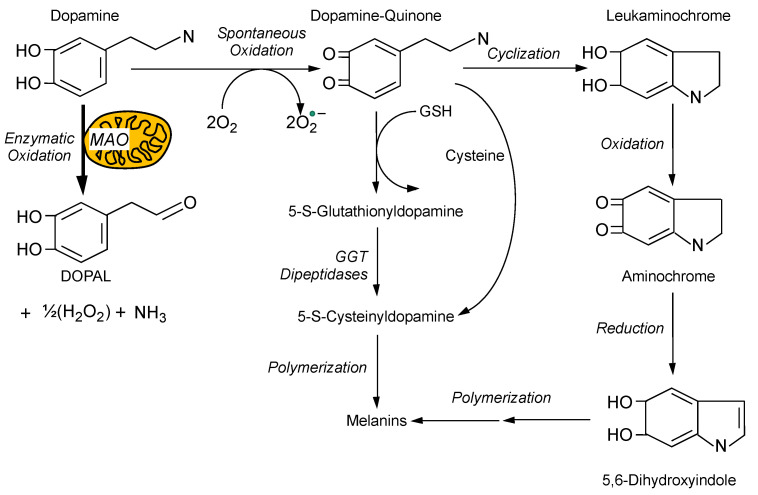
Enzymatic and spontaneous oxidation of dopamine. Monoamine oxidase (MAO) in the outer mitochondrial membrane converts dopamine to 3,4-dihydroxyphenylacetaldehyde (DOPAL), ammonia, and hydrogen peroxide. Dopamine oxidizes spontaneously to dopamine quinone, generating superoxide radical. Dopamine quinone can react SH) to form glutathionyldopamine, which, in the presence of gammaglutamyl transferase (GGT) and dipeptidases, results in the formation of 5-S-cysteinyldopamine (Cys-DA). Cys-DA can also be produced by dopamine quinone reacting with cysteine. Cyclization of dopamine quinone produces leukaminochrome, which can be oxidized to form aminochrome. Reduction of aminochrome yields the indole 5,6-dihydroxyindole. Melanins are formed from polymerization of Cys-DA or 5,6-dihydroxyindole. The relatively thick arrow corresponding to MAO is used to denote that given the alternative spontaneous vs. enzyme-catalyzed oxidation of cytoplasmic dopamine, the enzymatic route would predominate; however, MAO inhibition could increase the formation of spontaneous oxidation products.

**Figure 2 ijms-22-05999-f002:**
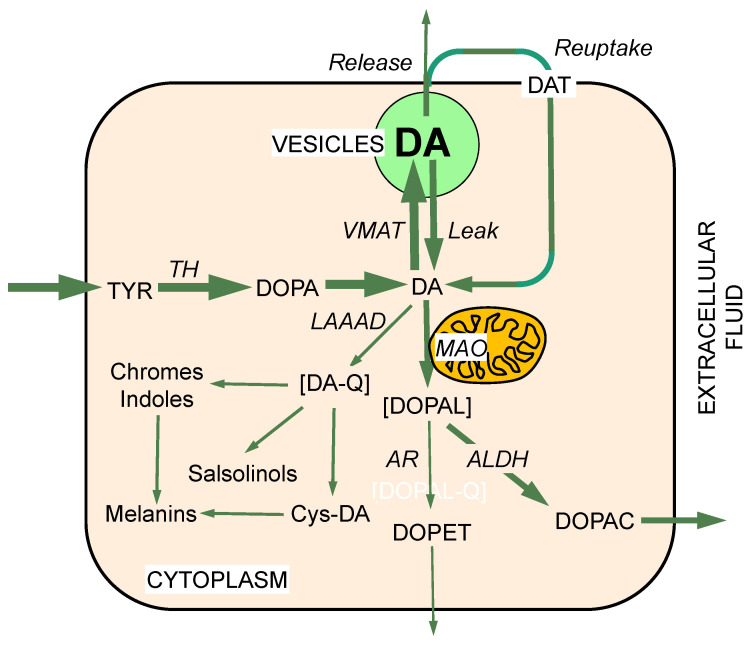
Dopamine synthesis, storage, reuptake, and metabolism. Dopamine (DA) is produced in the neuronal cytoplasm via tyrosine hydroxylase (TH) acting on tyrosine to form 3,4-dihydroxyphenylalanine (DOPA) and then L-aromatic-amino-acid decarboxylase (LAAAD) acting on DOPA to form dopamine. Most of cytoplasmic DA is taken up into vesicles by way of the vesicular monoamine transporter (VMAT). This is balanced by passive leakage from the vesicles into the cytoplasm. Cytoplasmic DA is subject to oxidative deamination catalyzed by monoamine oxidase-A (MAO-A) in the outer mitochondrial membrane to form 3,4-dihydroxyphenylacetaldehyde (DOPAL). DOPAL is converted to 3,4-dihydroxyphenylacetic acid (DOPAC) via aldehyde dehydrogenase (ALDH). A minor pathway is via aldehyde/aldose reductase (AR) to form 3,4-dihydroxyphenylethanol (DOPET). Most of vesicular DA released by exocytosis is taken back up into the cytoplasm via the cell membrane dopamine transporter (DAT). DA can undergo spontaneous oxidation to DA quinone (DA-Q), resulting in the formation of a variety of oxidation products including 5-S-cysteinyldopamine (Cys-DA).

**Figure 3 ijms-22-05999-f003:**
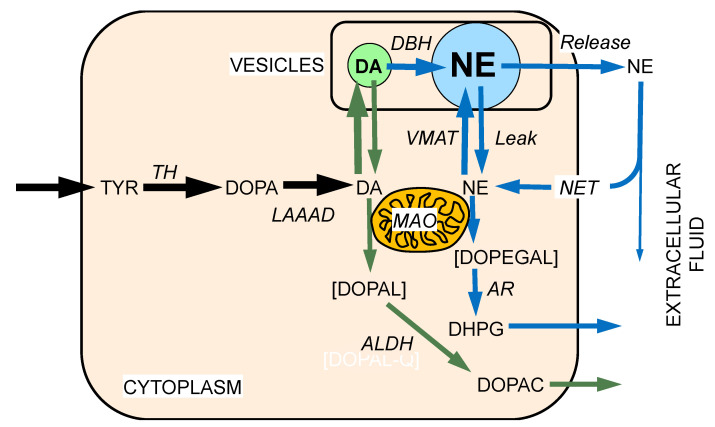
Norepinephrine synthesis, storage, reuptake, and metabolism. In noradrenergic neurons, dopamine (DA) is produced in the cytoplasm as in dopaminergic neurons. Dopamine-beta-hydroxylase (DBH) in the vesicles catalyzes the production of norepinephrine (NE) from DA. As for DA, cytoplasmic NE is subject to oxidative deamination catalyzed by monoamine oxidase-A (MAO-A) in the outer mitochondrial membrane, to form 3,4-dihydroxyphenylglycolaldehyde (DOPEGAL). It should be noted that NE is a poorer substrate than DA for MAO of either isoform. DOPEGAL is converted to 3,4-dihydroxyphenylglycol (DHPG) via aldehyde/aldose reductase (AR). A minor pathway (not shown) involves conversion of DOPEGAL to dihydroxymandelic acid via aldehyde dehydrogenase (ALDH). As for DA, most of vesicular NE released by exocytosis is taken back up into the cytoplasm via a cell membrane transporter—the cell membrane norepinephrine transporter (NET) for NE (although DA is a better substrate than NE for uptake via the NET). Theoretically, NE can be oxidized spontaneously to NE quinone and downstream oxidation products. Other abbreviations: LAAAD = L-aromatic-amino-acid decarboxylase; TH = tyrosine hydroxylase; TYR = tyrosine; VMAT = vesicular monoamine transporter.

**Figure 4 ijms-22-05999-f004:**
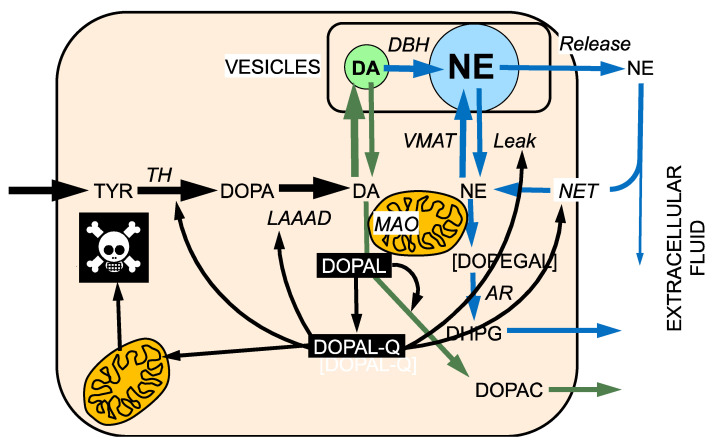
Intraneuronal effects of DOPAL buildup. DOPAL buildup in the neuronal cytoplasm may exert numerous harmful effects. These can be classified in terms of cytotoxicity via adverse effects on mitochondria and abnormalities from protein modifications, probably mediated by DOPAL quinone (DOPAL-Q). For instance, DOPAL inhibits tyrosine hydroxylase (TH) and L-aromatic-amino-acid decarboxylase (LAAAD). DOPAL can interfere its own detoxification by inhibiting aldehyde dehydrogenase (ALDH). DOPAL also decreases the storage of catecholamines in vesicles, by inhibiting the cell membrane norepinephrine transporter (NET) or the vesicular monoamine transporter (VMAT) or permeabilizing vesicles.

**Figure 5 ijms-22-05999-f005:**
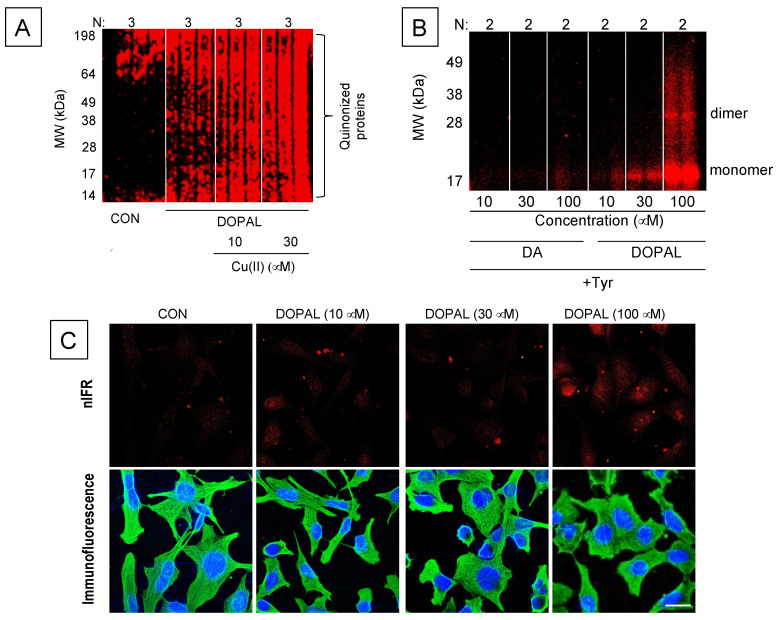
Intracellular quinoproteins produced by 3,4-dihydroxyphenylacetaldehyde (DOPAL)**.** (**A**): DOPAL-induced quinonization of intracellular proteins in MO3.13 human cultured oligodendrocytes. MO3.13 cells (1.5 × 10^5^ cells/well) were exposed to (**A**) DOPAL (100 μM) with or without Cu(II) (10 and 30 μM) for 24 h. DOPAL-quinonized proteins were detected and quantified by infrared fluorescence (nIRF) spectroscopy (red). N = number of replicates. DOPAL quinonized myriad intracellular proteins. Cu(II) augmented this effect. (**B**): Comparisons of dopamine (DA)- vs. DOPAL-induced alpha-synuclein (αS) quinonization. Quinonized αS was detected by near infrared fluorescence (nIRF) spectroscopy (red). Even in the presence of tyrosinase (Tyr) to oxidize DA, DA did not quinonize αS. (**C**): Visualization of intracellular DOPAL-induced quinoproteins. MO3.13 cells were cultured in slide chambers (8 × 10^4^ cells/slides) for 24 h and treated with Cu(II) (30 μM) and 0–100 μM DOPAL for 5 h. Cells were then stained with DAPI (1:2000) (blue) and human tubulin antibody (1:1500) (green). Immunofluorescence and near infrared fluorescence (nIRF) were visualized microscopically. Scale bar in images is 20 μM. Treatment with DOPAL produced nIRF signals, indicating the presence of quinoproteins. Images are from [122].

**Figure 6 ijms-22-05999-f006:**
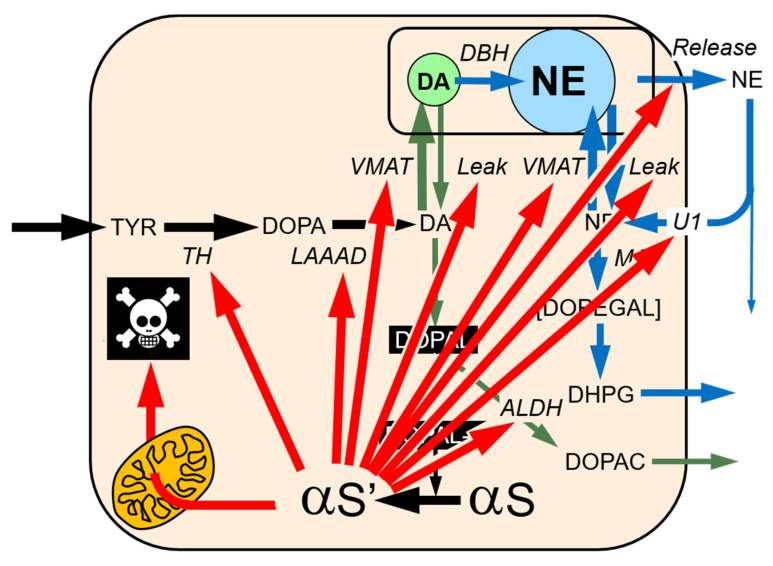
Some effects of 3,4-dihydroxyphenylacetaldehyde (DOPAL)-induced modification of alpha-synuclein (αS). Among myriad protein modifications exerted by DOPAL, one of the most prominent is oligomerization of, formation of quinone–protein adducts with (“quinonization”), and aggregation of αS. DOPAL-induced αS oligomers interfere with vesicular functions and are toxic to mitochondria. Not shown, αS may increase DOPAL formation via stimulation of MAO activity. The relative roles of DOPAL alone, αS alone, and their interactions in catecholaminergic neurodegeneration is poorly understood. It is also possible that αS and DOPAL emanating from catecholaminergic neurons can be taken up into non-neuronal glial cells and harm the neurons indirectly via altered glial functions.

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
