# Peer review of "The Catecholaldehyde Hypothesis for the Pathogenesis of Catecholaminergic Neurodegeneration: What We Know and What We Do Not Know"

_ijms, 2021, doi:10.3390/ijms22115999_

Round 1

Reviewer 1 Report

The review of Goldstein provides a comprehensive picture for the proposed role of catecholaldehydes in neurodegeneration. The focus is the dopamine metabolite 3,4-dihydroxyphenylacetaldehyde (DOPAL), however, available details for 3,4-dihydroxyphenylglycolaldehyde, the norepinephrine metabolite, are discussed as well. Past as well as very recent evidence is presented along with excellent interpretation and analysis. Overall, the manuscript is well-written, adequately detailed and of much interest (or should be) to those in fields relevant to the primary topic. Only minor suggestions are provided.

Line 224. Note that the ALDH enzymes are also inhibited by products of oxidative stress (lipid peroxidation products).

Line 301. The papers cited present evidence that COX-2 can also oxidize DOPAL to the unstable quinone (or perhaps initially to the semiquinone radical), however, the relevance and physiologic consequence of this is unknown.

Line 361. Also note that the reaction of DOPAL with Lys residues produces reactive oxygen species, likely via an oxidative rearrangement. NAC (as well as GSH and Ascorbate) can mitigate protein modification, likely via preventing this oxidative step (Line 475).

Line 250. Yes, it appears the Graves model assumes DOPAL oxidation via ALDH is cytosolic (ALDH1), however, mitochondrial ALDH (ALDH2) is known to metabolize DOPAL and would generate mitochondrial NADH.

Author Response

Please see the attached for detailed.

Reviewer 2 Report

The manuscript covers the authors' theory on a role of DOPAL in the pathogenesis of Parkinson's disease. This is well-written review which gives the almost complete understandings on the mechanisms of DOPAL generation, conversion and toxicity. The main weak point of the manuscript is the figures (schemes). They need to be redrawn to avoid numerous crossing of lines and arrows, to visualize the most critical points of the theory. Also, the authors should detalize their notes on gut-brain interactions in the context of peripherally-induced synuclein pathology and spreading. I also recommend to pay more attention to the possibility to apply the data to the early (at the pre-motor stage) diagnostics or prevention of PD. Authors should discuss whether the same mechanism could explain PD-associated changes in brain plasticity (i.e. neurogenesis) and development of neuroinflammation.

Author Response

Please see the attached for detailed.

Round 2

Reviewer 1 Report

Authors adequately addressed my comments

Reviewer 2 Report

Revised version of the manuscript is improved.